# Prospects and Future Directions of Self-Healing Fiber-Reinforced Composite Materials

**DOI:** 10.3390/polym12020379

**Published:** 2020-02-08

**Authors:** Min Wook Lee

**Affiliations:** Institute of Advanced Composite Materials, Korea Institute of Science and Technology, Chudong-ro, Bongdong-eub, Jeonbuk 55324, Korea; mwlee0713@kist.re.kr

**Keywords:** self-healing, composite, structural materials, extreme conditions

## Abstract

In this paper, the anticipated challenges and future applications of self-healing composite materials are outlined. The progress made, from the classical literature to the most recent approaches, is summarized as follows: general history of current self-healing engineering materials, self-healing of structural composite materials, and self-healing under extreme conditions. Finally, the next stage of research on self-healing composites is discussed.

## 1. Introduction

In nature, self-healing is an autonomous and a fascinating phenomenon that can be observed in most living organisms. Wounds such as scratched skin or fractured bones are easily healed through the activity of the vascular system [1]. The survival of animals and plants depends on their expansion owing to their restorative capacity. Such bio-inspired recovery of engineering materials, namely, "self-healing" following external damage, has been studied for the past 30 years [2,3]. When blood is released from scratched skin, healing substances are released, solidify, and aggregate along the ruptured area. Nature-inspired self-healing features have been explored in biomimetic designs and healing strategies. Structural damage is repaired through the systematic transportation of the healed material and the polymerization-based curing process of the damaged area. The first generation of self-healing studies were performed using microcapsules [4]. The capsule was certainly viable and did not require external energy to begin the healing process. However, the layer with the capsule was thick owing to the bulky microcapsules. Moreover, in the view of repeatable healing, these capsules were used only once. Therefore, new approaches for small healing substances and multiple healing abilities were required.

As observed in mammals or plants, the vascular network enables rapid and continuous transportation of healing substances to the damaged area. This effective microvascular system is composed of a network structure and perfectly covers the entire body/surface. For the Gen. II self-healing research, the main mechanism of such self-healing, the vascular capillary network, which carries the healing substance, was studied [5,6,7].

Although numerous self-healing materials have been developed, it remains unclear whether the methods proposed for fabricating these materials are economically feasible and scalable to the industrial level. For example, capsule-based self-healing methods exhibit several disadvantages, such as low uniformity of the dispersed capsules and complicated fabrication processes [8]. To overcome these drawbacks, several fiber-based self-healing approaches have been introduced in recent years, one of which, solution blowing, has already been scaled up. Nevertheless, the range of materials that can be used as a shell for encasing the core materials in these core-shell nanofibers is severely limited. Capsule-based methods allow for the use of a wider range of materials within their limitations. 

Moreover, hybrid methods, including both capsule-based and fiber-based approaches, require further development to maximize their advantages. For example, self-healing composites consisting of fast-healing capsules, and small-sized self-healing core-shell nanofibers can be used to repair damage in a wide range of cracks, such as those that are several nanometers in size (owing to the nanofibers). Moreover, thanks to capsules that are hundreds of nanometers wide, such a hybrid approach will not exhibit the limitations associated with slow healing in terms of low uniformity owing to the capsule use and the presence of nanofibers. Finally, the addition of corrosion inhibitors or the use of pH and redox polymers can further improve the self-healing performance [9]. Applications including the interfacial strengthening of composite laminates for the aviation and automotive industries remain significant as a means of protection against impact damage and fatigue cracking. Therefore, a nano-textured self-healing interleaved structure aimed at interfacial enhancement is required. Furthermore, the development of innovative technologies, such as the proliferation of soft robots and actuators, as well as products based thereon, requires innovative flexible self-healing composites that can withstand multiple operation cycles without fatigue crack growth. 

## 2. Self-Healing of Structural Composite Materials

Fiber-reinforced composites (FRCs) have received increasing attention owing to the demand for lightweight construction and high strength in construction and structural materials [10]. In particular, as high-performance composite materials consisting of glass fiber, aramid fiber and carbon fiber have been applied to large structures such as aircraft, the price competitiveness has increased, and their use has rapidly expanded over the past decade. The global composites market, which was valued at $90 billion in 2019, is expected to grow to $113.6 billion by 2024 with the replacement of many commonly used products, including sporting goods, cars, and protective gear [11]. However, items that are vulnerable to impact damage or repeated mechanical and thermal loading remain a major challenge. In particular, large structures that are exposed to constant vibration and shock for lengthy periods may be at risk of fatal failures from small cracks or defects. In 1996 and 1989, large accidents were reportedly caused by cracks in the aircraft fuselage [12] and nuclear power plant walls [13], respectively. Structural health assessments and maintenance costs are expected to reach $5.5 billion by 2025 to prevent major casualties or economic losses [14]. A lack of technical expertise and complications associated with the installation of structural health monitoring systems for large structures are factors hampering market growth. Structural health monitoring technology deals with the implementation of systems and techniques that include inspection, monitoring, and other processes for damage detection in structures. For this purpose, various studies have been conducted to provide self-healing functions to structural composites.

The repair of carbon fiber-reinforced polymer (CFRP) composites was first reported by Kessler et al. in 2003 [15]. In his study, “Self-healing structural composite materials”, microcapsules were loaded with healing materials to prevent delamination fracture of CFRP composites. The healing agent, dicyclopentadiene-encapsulated microcapsule (d = 166 μm) was mixed with an epoxy resin by 20 wt %. The healing agent was released from the broken capsules into the crack plane of the double cantilever beam. The inter-laminar fracture toughness of the specimen was recovered to 40% and 80% at room temperature and 80 °C, respectively. Another approach for recovery from delamination damage was studied using a thermoplastic polymer matrix [16]. The thermally responsive polyurethane that incorporated the Diels–Alder (DA) reaction achieved repeated healing of the delamination inside a carbon fiber composite with 85% and 75% healing efficiency during the first and second cycles, respectively. The damaged specimen was hot-pressed under 100 psi (the same as the fabrication pressure) and heated at 135 °C for 2 h, 90 °C for 2 h, and 70 °C for 2 h in series. Despite the low mechanical strength of the matrix material and the unavoidable use of the hot press and additional heating, the first intrinsic healing of CFRP matrix materials was successfully demonstrated. D’Elia et al. [17] investigated a brick-and-mortar structure using glass and poly(borosiloxane). The stiff block and supramolecular polymer provided mechanical strength and a repeatable healing function to the material, thereby achieving the design of a structural composite model with thermoplastic characteristics but high strength. The successful healing that enabled the structure to fully recover its original strength in ~MPa repeatedly was demonstrated under room temperature conditions. 

Thus far, the self-healing function has been provided by polymers in the matrix material in composites [18,19]. However, in reality, reinforcing fibers are responsible for the majority of the strength in composites. The recovery of these key elements, such as reconnection of cut carbon fibers, is impossible at this point. However, repair of the matrix material without considering the reinforcing fibers only addresses a small part of the problem. Efforts must be made to face and proactively focus on these fundamental and inevitable problems.

## 3. Self-Healing Under Extreme Conditions 

Although the majority of plants and animals live in suitable habitats, special creatures exist that adapt and live in extreme environments, for example, under the deep sea, deserts, tundra or polar regions. To broaden the range of human activities and enhance survivability, these methods of nature should be studied. 

### 3.1. Heat Generation 

Most plants and animals need to maintain an appropriate temperature range to facilitate enzyme activity and metabolism in the body [20]. This is also applicable to the healing systems of composite materials. In particular, polymer materials under curing conditions that are significantly affected by the temperature range require the exothermic function to ensure healing efficiency in a low-temperature environment. In general, most polymeric materials used as healing agents can only achieve successful curing within a narrow temperature window. When the moisture or pH effect is not taken into account, the reactivity of polymers is significantly reduced in a low-temperature environment [21,22,23]. 

Wang et al. [24] embedded an additional heating layer in a glass fiber-reinforced polymer composite to maintain the temperature of the healing system at an ultra-low temperature (−60 °C). The inter-laminar adhesion of plies was recovered by successfully releasing the healing agent with internal heating. Although the method of using the heating layer is simple, it is highly advantageous because it can operate in a low-temperature environment while using the existing polymer material. Most of the intrinsic healing systems thermally reversible bonding or thermoplastic epoxies are efficiently re-mended at elevated temperature [25]. For this reason, local heating is effective in increasing the recovering rate of shape memory polymers or supramolecular polymers [26].

Joule heating of conductive film was studied by Lee et al. [23]. An electroplated Cu nanofiber web, which was electrically conductive and transparent, was facilitated as a thin film heater alongside a bromobutyl rubber (BIIR) layer. The temperature of the Cu fiber film was increased to 150 °C at 3.6 V and 0.7 A, which was sufficient to mend the cracks and protect the steel substrate from corrosion repeatedly. In another publication from the same group, a carbon nanotube-embedded Polydimethyl siloxane (PDMS) or BIIR layer (the healing agent) was used as a soft film heater [21,27]. The heater aided in accelerating the release and polymerization of the PDMS, thereby reducing the healing process from 24 h to only 10 min. Effective healing has also been demonstrated under ice water bath (~1 °C) or salt water (4 wt %) conditions in simulations of marine structures exposed to the Arctic Ocean. These findings have practical significance, as typical polymer types of healing agents have a higher chance of survival in harsh environments simply with the aid of a thin film heater. This extends the efficiency and lifetime of self-healing applications in mechanical/chemical damage repair. 

### 3.2. New Materials and Approaches

In terms of reversibility, the majority of healing substances are classified as thermoplastic. A mendable thermoset polymer with a glass transition temperature (T_g_) of 270 °C and decomposition temperature (T_d_) of 365 °C was synthesized by Zhang et al. [28]. The isocyanurate-oxazolidone polymer incorporated with mechano-responsive isocyanurate rings recovers its broken bonding by means of thermal annealing. This novel mending chemistry is promising because such structures are thermoset but mendable. Most of the intrinsic healing materials are thermoplastic including the DA reaction mechanism. These materials exhibit excellent healing performance in terms of repeatability and high efficiency but also suffer from the critical deficiency of low mechanical strength compared to engineering plastics. This has been identified as the main factor limiting the self-healing applications of structural composite materials. Thermal stability at high temperatures is as important as low-temperature cleavage for self-healing polymers and composites. Heo et al. [29] reported that a self-healing polymer based on the DA reaction and its composite achieved a 94% and 69% healing efficiency, respectively, while the polymer was stable at 240 °C. Effective thermal stability with mechanical strength that is comparable to other engineering polymers and structural composites will be beneficial for the future of self-healing composite materials. 

In 2014, White et al. [30] demonstrated the restoration of not only a partial or moderate level of damage but also a permanent large volume of damage. A serious loss of material in the form of a punctured hole (diameter = 35 mm) was restored with thermoplastic gels supplied by the vascular delivery system. This result resembles the amazing restoration that regenerates the body of sea cucumbers or lizards. This level of damage and recovery indicates a self-healing function that will be required in the next generation. An improved approach based on smart ideas and the development of healing agents with superior physical properties will accelerate such developments.

### 3.3. Field Repair

The military standards or military specifications were originally proposed as an evaluation standard for military products, but in recent years, these have also been used to provide a high level of durability and reliability in general products. These standards have been recognized as a performance index for operation under tough conditions, in which a field test is a mandatory requirement. Where no extensive facilities or specialized equipment are available, and without using hazardous materials, a certain level of performance should be achieved. Because the application of fiber-reinforced polymer composite materials is continually expanding in the army service area, including aviation (drones, helicopters), armor (vehicles and personal gear), ground vehicles, and tactical structures, repairability should be ensured as much as superior performance [31]. 

For this reason, the out-of-autoclave process is a recent trend at the forefront of composite manufacturing technology [32]. Moreover, the solution process or non-vacuum process is a promising approach that can be applied by means of simple and inexpensive techniques. Such methods are highly beneficial as the healing materials can be prepared in the form of solution, which is easy to handle and can be applied to any irregular surface shape or can be used to patch a damaged site. It would be advantageous for intrinsic healing materials, such as supramolecular polymers or rubbers, to be applied directly to the substrate to provide a protective film in real time. For example, ships and offshore constructions are constantly exposed to a corrosive marine environment consisting of salt water, rain, UV rays, and aquatic lives. To ensure human safety, the infrastructures and vehicles have to be maintained for protection against these unfavorable conditions. A BIIR (dissolved in hexane) solution was successfully coated on a corrosive steel substrate by means of casting, air spraying and brush-painting [27]. The self-healing film could mend cracks and protect the substrate against 4 wt % saltwater in 2 h. This is one promising means of providing field repair to structural materials.

## 4. Concluding Remarks

In addition to the cracks on aircraft fuselage and nuclear power plant walls mentioned above, numerous structures require protection as human activities expand to such areas as submarine tunnels and spacecraft. As the space age accelerates, it is expected that individuals will actually travel through space and explore unknown worlds. For example, as events such as a micro-meteor impacting personal spacesuits at 50 km/s [33] or piles of stones hitting undersea tunnels for years no longer exist only in the imagination, the performance and circumstances of self-healing in the future should be considered under more severe conditions. Very fortunately, for the past 30 years, materials and successful approaches have been found that enlarge self-healing applications. Hopefully, the core principles will be understood soon, and the fundamental solutions will be revealed.

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
