# Peer review of "Prospects and Future Directions of Self-Healing Fiber-Reinforced Composite Materials"

_polymers, 2020, doi:10.3390/polym12020379_

Round 1

Reviewer 1 Report

The manuscript entitled “Prospects and future directions of self-healing composite materials” critically and objectively gathers the latest advances made in the field of self-repairable composite materials.

The work is very well structured and written. The presentation of the results and their critical interpretation is clear and concise.For all these reasons, I consider that this work can be published directly in a high-impact journal such as Polymers, without mentioning any profound changes.

As a small change, indicate that on page 4, line 149 the temperature ranges are not defined, only the ## symbols appear. Please correct.

Author Response

We thank the Reviewer for the comments and suggestions. Summarized below are our responses to the Reviewer’s specific concerns.

Comment 1:   As a small change, indicate that on page 4, line 149 the temperature ranges are not defined, only the ## symbols appear. Please correct.

Response 1: The temperature condition was mentioned as 240 °C and typo was removed in the sentence on p. 8.

Reviewer 2 Report

The authors have presented a very well written manuscript. It is brief but comprehensive. I think that the authors should provide additional references in some parts of the text in order to enrich it. Also some statements must have the specific reference (see pdf attached).

Finally the title must be changed. The authors have a very general title "....self-healing composite materials", where in the introduction and in the text they do not cover all the aspects of self-healing materials. Furthermore they are studying specific areas of applications and are concentrated in capsules and fibers. Therefore, I believe the title should be more concentrated to the materials refered in the text in order to help the readers interested for these kind of materials/applications to read it.

Author Response

We thank the Reviewer for the comments and suggestions. Summarized below are our responses to the Reviewer’s specific concerns.

Comment 1:   Lines 17-25: Provide references for all these statements

Response 1: Ref. 1 and 2-3 were added in the sentences.

Comment 2:   Lines 38-39: Reference

Response 2: Ref. 8 was added in the sentences.

Comment 3:   Lines 51-52: Reference

Response 3: Ref. 9 was added in the sentences.

Comment 4:   From the introduction, the author clearly is aiming at specific areas of application for self-healing materials and is focusing to capsules and fibers. Therefore, I believe that the title of the manuscript must change and be more accurate for these type of materials. The title is too general “…self-healing composite materials” and it does not reflect to the whole literature of the self-healing composite materials.

Response 4: The title was changed to specify the topic as ‘Prospects and future directions of self-healing fiber-reinforced composite materials’.

Comment 5:   Lines 60-61: Reference

Response 5: Ref. 10 was added in the sentences.

Comment 6:   Lines 64-66: Reference

Response 6: Ref. 11 was added in the sentences.

Comment 7:   Lines 71-72: Reference

Response 7: Ref. 14 was added in the sentences.

Comment 8:   Lines 97-98: Reference

Response 8: Ref. 18-19 were added in the sentences.

Comment 9:   Lines 114-117: Reference

Response 9: Ref. 21 was added in the sentences.

Comment 10: 3.1: I think the authors should provide more references for these type of polymers

Response 10: Ref. 20, 21-23, 25, 26 were added in the sentences.

Comment 11: Line 149: Please correct “at ## and ##”

Response 11: The temperature condition was mentioned as 240 °C and typo was removed in the sentence on p. 8.

Comment 12: Lines 173-177: Reference

Response 12: Ref. 23 was added in the sentences.

Comment 13: Line 178: Define “BIIR” before using its abbreviation

Response 13: In line 125, ‘bromobutyl rubber (BIIR) ~’ was mentioned at the first statement.
